# Modulating the Heat Sensitivity of Prostate Cancer Cell Lines In Vitro: A New Impact for Focal Therapies

**DOI:** 10.3390/biomedicines8120585

**Published:** 2020-12-09

**Authors:** Oliver Hahn, Franziska M. Heining, Jörn Janzen, Johanna C. R. Becker, Marina Bertlich, Paul Thelen, Josef J. Mansour, Stefan Duensing, Sascha Pahernik, Lutz Trojan, Ionel V. Popeneciu

**Affiliations:** 1Department of Urology, University Medical Center Göttingen, 37075 Göttingen, Germany; franziska-heining@gmx.de (F.M.H.); joerntoepperwien23@gmail.com (J.J.); j.becker01@stud.uni-goettingen.de (J.C.R.B.); marina.bertlich@med.uni-goettingen.de (M.B.); marion.striepe@med.uni-goettingen.de (P.T.); lutz.trojan@med.uni-goettingen.de (L.T.); 2Department of Urology, Heidelberg School of Medicine, University of Heidelberg, 69120 Heidelberg, Germany; josef.mansour@klinikum-karlsruhe.de (J.J.M.); stefan.duensing@med.uni-heidelberg.de (S.D.); sascha.pahernik@klinikum-nuernberg.de (S.P.); 3Department of Urology, Paracelsus Medical University Nuremberg, 90419 Nuremberg, Germany

**Keywords:** prostate cancer, focal therapy, HiFu, thermosensitivity, androgen deprivation

## Abstract

Focal therapies such as high-intensity focused ultrasound (HiFU) are an emerging therapeutic option for prostate cancer (PCA). Thermal or mechanical effects mediate most therapies. Moreover, locally administered drugs such as bicalutamide or docetaxel are new focal therapeutic options. We assessed the impact of such focal medical treatments on cell viability and heat sensitivity by pre-treating PCA cell lines and then gradually exposing them to heat. The individual heat response of the cell lines tested differed largely. Vertebral-Cancer of the Prostate (VCaP) cells showed an increase in metabolic activity at 40–50 °C. Androgen receptor (AR)-negative PC3 cells showed an increase at 51.3 °C and were overall more resistant to higher temperatures. Pre-treatment of VCaP cells with testosterone (VCaPrev) leads to a more PC3-like kinetic of the heat response. Pre-treatment with finasteride and bicalutamide did not cause changes in heat sensitivity in any cell line. Mitoxantrone treatment, however, shifted heat-induced proliferation loss to lower temperature in VCaP cells. Further analysis via RNAseq identified a possible correlation of heat resistance with H3K27me3-dependent gene regulation, which could be related to an increase in the histone methyltransferase EZH2 and a possible neuroendocrine differentiation. Pre-treatment with mitoxantrone might be a perspective for HiFU treatment. Further studies are needed to evaluate possible combinations with Hsp90 or EZH2 inhibitors.

## 1. Introduction

Prostate cancer (PCA) is the most common malignant disease in men in 2020 with a predicted incidence of 191,930 new cases in the United States [1]. Treatment options in localized stages are surgery and radiotherapy as well as Active Surveillance (AS) for low-risk tumors. In recent studies, it has been shown, however, that, in men undergoing observation as primary treatment for intermediate-risk tumors, disease progression and additional treatment are imminent [2]. To escape this dilemma, an emerging therapeutic option especially for low-volume or localized and lower risk disease are local, subtotal or focal therapies such as high-intensity focused ultrasound (HiFU), TOOKAD (photodynamic therapy with Pd-bacteriopheophorbide), or cryotherapy [3]. Focal therapies are widely used in other malignancies such as hepatocellular carcinoma, renal cell carcinoma or thyroid carcinoma. Through different mechanisms, destruction of the tissue surrounding an energy source (e.g., an ultrasound probe) induces local tissue damage and thereby destroys the tumor. A basic principle is the treatment of the index lesion which is defined as the lesion with the highest grade of malignancy [4].

The effect of HiFU is achieved by (a) coagulative cell death caused by high temperatures and (b) mechanical cavitation caused by the effect of the ultrasound [5]. The temperature reached in the tumor core after the application of HiFU reaches around 60 °C and is held for up to 20 s, thereby inducing coagulative necrosis and immediate cell death [6,7,8,9]. In vitro evidence suggests temperatures above 55 °C and an exposure time of one second to be sufficient [7,10].

Heat distribution around the focus of the ultrasound probe decreases concentrically. In order to get temperatures of at least 55 °C in all areas of the tumor, the core needs to reach >90 °C. At lower temperatures, there has been detection of viable tumor cells specifically at the margin of the treatment zone. Irreversible cell damage in these areas occurs only after prolonged exposure (up to 60 min) [11]. One of the aims of this study was therefore to validate this finding in vitro and to determine at which temperature PCA cell lines are definitely destroyed in an optimal intensity/duration ratio.

Focal chemotherapy has become an emerging alternative to systemic chemotherapy. Using conjugation of therapeutics to polymers such as polyglutamic acid (PGA), these drugs can be held in the tumor using the so-called enhanced permeability and retention (EPR) effect. Coupled to a high molecular-weight nanocarrier, the drugs cannot extravasate anywhere but in the leaky vessels of the tumor tissue [12]. Such conjugates have successfully been tested in vivo for focal therapy of hepatoma and showed promising results [13]. For prostate cancer, ST-4PC, a polymeric paste containing docetaxel and bicalutamide, has been shown to work in vivo for focal ablation of orthotopic tumors in mice [14]. Therefore, our second aim was to investigate the effect of these drugs (as well as other anti-androgenic and chemotherapeutic agents) on heat sensitivity in order to lay the groundwork for possible combinatorial approaches with classical HiFU therapy.

## 2. Materials and Methods

### 2.1. Cell Culture and Pre-Treatment

Prostate cancer cell lines Vertebral-Cancer of the Prostate (VCaP), Prostatic carcinema cell line 3 (PC-3), 22rv1 and Benign Prostate hyperplasia 1 (BPH-1) were purchased from LGC Standards (Teddington, UK) or used from local certified stocks as previously described [15,16]. Briefly, cells were cultured in phenol-red free Dulbecco’s Modified Eagle’s Medium (DMEM) with 10% fetal bovine serum (both Sigma Aldrich, Taufkirchen, Germany). For the experiments, cells were kept under 35 passages and confluence was kept under 80%. Androgen-dependent VCaPrev cells were kept with 1 nM Testosterone over 20 passages as described before [17]. Cells were incubated at 37 °C and 5% CO_2_. For treatment, cells were seeded in a 6-well-plate and cultivated for 48 h, aiming at a quantity of 1 × 10^6^ cells for further analysis. After 48 h, cells were treated with bicalutamide (Sigma-Aldrich), finasteride (Sigma-Aldrich), mitoxantrone (Hexal, Holzkirchen, Germany), or docetaxel (Amneal, Gröbenzell, Germany). As a control, DMSO or ethanol (both Sigma-Aldrich) were used in equal amounts. Cells were then harvested after an additional 48 h and used for further analysis.

### 2.2. Heat Stress Induction

After being transferred into PCR tubes, cells were gradually heated in a thermal cycler (C1000, BioRad, Hercules, CA, USA) with a ramping rate of 3.5 °C/s and 5 s pre-exposure at 37 °C. Exposing temperatures ranked between 37 and 100 °C, and the final temperature was kept for 10 s before cooling down again. Afterwards, cells were used for further analysis.

### 2.3. Measurement of Metabolic Activity

Metabolic activity was measured using the CellTiter 96 AQueous One Solution Cell Proliferation Assay (Promega, Madison, WI, USA) according to the manufacturer’s instructions. Briefly, 77 µL of cell-specific media was added to 33 µL of cell suspension. After transferring the mixture into a 96-well-plate, 20 µL of 3-(4,5-dimethylthiazol-2-yl)-2,5-diphenyltetrazolium bromide (MTT) stain was added before incubating the plate for 4 h. Afterwards, the absorption of the plate at 490 nm was read in a M200 PRO reader (Tecan, Männedorf, Switzerland) and measured relative to a medium control. Afterwards, results were analyzed and displayed relative to a 37 °C control for each treatment condition. A medium control without cells was subtracted from each absorbance value measured before analysis.

### 2.4. PI/Annexin V Staining and Flow Cytometry

To further characterize cell death after thermal stress, we used the Fluorescein isothiocyanate (FITC) Annexin V Apoptosis Detection Kit II (BD Biosciences, San Jose, CA, USA). Immediately afterwards, cells were analyzed in a FACS (fluorescence-activated cell sorting) Canto II Flow Cytometer (BD Biosciences) at the FACS core facility of the University Medical Center Göttingen (UMG). For each condition, 10,000 cells were counted. Analysis was done using FlowJo Single Cell Analysis Software v10 (FlowJo LLC, Ashland, OR, USA).

### 2.5. RNA Sequencing (RNAseq)

RNA was extracted using phenol-chloroform extraction following a standard protocol. RNA integrity was checked on a 1.5% Agarose gel. Afterwards, library preparation was done with a total amount of 1 µg RNA per sample using the TruSeq RNA Library Prep Kit v2 (Illumina Inc., San Diego, CA, USA). Size range was verified to be around 280 bp using BioAnalyzer 2100 (Agilent Technologies, Santa Clara, CA, USA). Sequencing was done at the NGS core facility (NIG) at the University Medical Center Göttingen on an Illumina HiSeq 4000 platform. The results were processed using the BaseCaller to bcl files function in the Illumina software. De-multiplexing was done using bcl2fastq (version 2.17.1.14). Fastq files were then mapped to the human transcriptome (UCSC HG19) using TopHat2 (version 2.1.0) [18] with very sensitive Bowtie2 settings. The Cuffdiff function of the Cufflinks package (version 2.2.1) was used to analyze differential gene expression [19]. Volcano plots were created on the Galaxy platform (version 20.01) of the Gesellschaft für wissenschaftliche Datenverarbeitung mbH Göttingen (galaxy.gwdg.de) using a significance threshold of 0.05 [20,21]. Gene Set Enrichment analysis (GSEA) was performed with standard parameters (1000 permutations for gene sets, Signal2Noise metric for ranking genes) and significantly enriched pathways (c2.all gene sets) were shown [22]. The RNAseq data were deposited in the Gene Expression Omnibus (GEO) database (accession number GSE159531).

### 2.6. Sequencing Validation

Sequencing results were validated using qRT-PCR. Total RNA was isolated from cells using the Quick-RNA™ MiniPrep Kit (Zymo Research, Irvine, CA, USA) according to the manufacturer’s instruction. A total of 1 µg of RNA was then used for reverse transcription using the OmniScript RT Kit (Quiagen, Hilden, Germany) with random nonamer primers (IBA Lifesciences, Göttingen, Germany). qPCR was performed on a CFX96 system using SsoAdvanced Universal SYBR Green Supermix (BioRad, Hercules, CA, USA). All targets were normalized to the housekeeping gene *GAPDH*. Primer sequences were as follows: *IGFBP3* forward 5′-GAACTTCTCCTCCGAGTCCA-3′, *IGFBP3* reverse 5′-CTGGGACTCAGCACATTGAG-3′; *RUNX2* forward 5′-TCGGAGAGGTACCAGATGGG-3′, *RUNX2* reverse 5′-CATTCCGGAGCTCAGCAGAA-3′; DLX 3 forward 5′-TTTTCACCTGTGTCTGCGTGA-3′ DLX3 reverse 5′-GAAGCCCAAGAAGGTC CGAA-3′; *GAPDH* forward 5′-ACATCGCTCAGACACCATG-3′, *GAPDH* reverse 5′-TGTAGTTGAGGTCAATGAAGGG-3′. Data were analyzed using GraphPad Prism, version 8.4.3 (GraphPad Software, San Diego, CA, USA).

For Western blot analysis, cells were trypsinized and resuspended in RIPA buffer (Thermo Fisher Scientific, Waltham, MA, USA) supplemented with 1:200 PMSF (Roche Holding AG, Basel, Switzerland). Cells were incubated for 30 min on ice and spun down prior to protein measurement with DC Protein Assay (BioRad, Hercules, CA, USA). Afterwards, equal amounts of protein were loaded for SDS-PAGE. Blotting was performed using the TurboBlot system (BioRad, Hercules, CA, USA). Membranes were afterwards blocked with 5% milk powder in TBS+0.1% Tween20 before incubation with the respective antibody dilutions at 4 °C over night. The following primary antibodies were used: anti-EZH2 (Cell Signaling, Denvers, MA, USA, #5246S), anti-GAPDH (Origene (Rockville, MD, USA) #TA802519), anti-HSC70 (Santa Cruz Biotechnology, Dallas, TX, USA, #sc-7298), anti-p53 (Santa Cruz Biotechnology #sc-126). The next day, membranes were incubated with secondary antibody (Dako Agilent, Santa Clara, CA, USA, #P0260/#P0448) for 1 h before detecting protein bands using a ChemiDoc MP Imagine System with Clarity Western ECL Substrate (BioRad, Hercules, CA, USA).

## 3. Results

### 3.1. Heat Response Differs Significantly between AR-Positive and AR-Negative Cell Lines

All PCA cell lines were treated with temperatures between 37 and 100 °C. After that, metabolic activity was measured as a correlate of viability (Figure 1a). We noticed an initial increase in viability in some cell lines, most prominently in the VCaP cells which showed an increase around 1.4x the original viability at temperatures between 40 and 50 °C. A similar increase, although much lower, could be noted in 22rv1 cells but not in BPH1 or PC3 cells. In PC3 cells, however, we noticed a minimal increase in cell viability at 51.3 °C. After 57 °C, metabolic activity slowly decreased in most cells. In PC3 cells, a significant decrease was only visible at 62.6 °C. To further characterize cell death during this heat response, we next did PI/Annexin V staining and FACS analysis. As metabolic activity can also be influenced by heat regardless of cellular viability, we also used this method as a verification of the results gained previously. As key temperatures, 37 °C was used as a control, 51 °C as a temperature at which we expected an increase in viability in VCaP and 22rv1 cells, 57 and 62 °C as hallmarks of the noted drop in vitality, and 92 °C as a positive control for heat-induced cell death. For PC3 cells, as the temperature curve differed slightly in Figure 1a, we used 37, 62.6, 65 and 92.7 °C. The results are shown in Figure 1b–f. The initial increase in the viability of VCaP and 22rv1 shown in the MTT assay did not have any influence on the apoptotic activity. This is most likely due to an initial increase in metabolic activity caused by the temperature optimum for the enzymatic activity in the cells. However, we noticed a much later decrease in the PI/Annexin V negative cells in BPH1 and PC3 cells as compared to VCaP and 22rv1. As the latter are androgen receptor (AR) negative cells, we next decided to investigate the influence of AR modulation on heat sensitivity.

### 3.2. Inhibition of the Androgen Axis Using Bicalutamide and Finasteride Does Not Affect Heat Sensitivity

In order to investigate the impact of the androgen axis on heat sensitivity, we next sought to test the impact of AR inhibition and 5α-Reductase inhibition on heat sensitivity of androgen-dependent prostate cancer cells. Therefore, VCaP and 22rv1 cells were treated with 0.1, 1, 5 and 10 µM of respective inhibitor. The solvent (ethanol or DMSO) was used as a control. The results of the MTT assay and FACS analysis are shown in Figure 2. The results show no significant difference in heat response between treated and untreated cells neither in metabolic activity nor regarding the apoptotic response measured via FACS analysis.

### 3.3. The VCaPrev Cell Model of Castration Sensitive Prostate Canwcer Shows a Higher Increase in Metabolic Activity Compared to the Other Cell Lines

To further investigate the relationship between the androgen axis and heat response, we next sought to investigate the heat response on a castration sensitive cell line. Therefore, we used the VCaPrev model described earlier by our group [17]. Briefly, VCaP cells were treated with supraphysiological concentrations of testosterone, thereby inducing a stage of hormone dependency again. These cells were then subjected to the same treatments as the other cell lines used before. Interestingly, upon visual inspection, the initial increase in metabolic activity was significantly higher in VCaPrev cells as compared to VCaP WT cells (Figure 3a). At 51.3 °C, a statistically significant difference was noted (Figure 3b). Treatment with bicalutamide and finasteride, however, did not affect this increase in viability, which indicates that the effect is not exclusively dependent on AR activity (Figure 3c,d).

The PI/Annexin V staining of VCaPrev cells after heating reflected this initial increase, as, at 57.8 °C, the cells showed a significantly higher percentage of viable cells compared to VCaP and 22rv1 cells (Figure 3e,f). As AR-negative cell lines, BPH1 and PC3, respectively, in general showed a decrease in metabolic activity only at higher temperatures, we also compared the amount of viable cells at 62.6 °C between PC3, BPH1 and VCaPrev cells (Figure 3g). Again, a significant difference (though not as pronounced as in AR-positive cells) could be noted between VCaP rev and the other cell lines.

### 3.4. Pre-Treatment with Mitoxantrone and Docetaxel Decreases Metabolic Activity but Does Not Increase Heat-Induced Cell Death

Previously, the activity of the DNA type II topoisomerase has been shown to increase with heat shock in cancer cells [23,24,25]. As mitoxantrone, a type II topoisomerase inhibitor, is frequently used in PCA treatment, we decided to repeat the experiments also with this drug (Figure 4a,b). To make sure a possible effect of mitoxantrone on heat sensitivity would not be mediated merely by its cytotoxic effect, we also used the PCA chemotherapeutic docetaxel as a control (Figure 4c,d). Of note, both docetaxel (50 or 100 nM) and mitoxantrone (2,5 or 5 µM) lead to an initial decrease in viability already at lower temperatures. Pre-treatment of VCaP cells with mitoxantrone and with docetaxel shifted the heat-mediated decrease in viability to significantly lower temperatures, though the shift was far more pronounced with mitoxantrone. At 51.3 °C temperature, a significant difference between controls and treated cells was notable (Figure 4e,f). In PC3 cells, no significant change in viability was visible. However, there was no significant induction of PI/Annexin V positivity visible at 57.8 °C (Figure 4g,h), indicating that the decrease in metabolic activity is not caused by apoptosis.

### 3.5. RNA Sequencing of VCaP and VCaPrev Reveals a Highly Significant Enrichment of Genes Normally Repressed by H3K27me3 in VCaPrev Cells

In order to investigate possible reasons for the difference in heat response between VCaP and VCaPrev cells, we next did RNA sequencing of these cell lines. In total, we identified 1569 genes differentially regulated between these cells (Figure 5a; false discovery rate (FDR) < 0.05, log2FC >1 or <−1). A total of 873 of these genes were enriched in VCaPrev cells and only 696 genes were enriched in VCaP WT (for a complete list see Appendix A). Of note, there was no difference in heat shock proteins such as Hsp90 or Hsp70 detectable. Next, we performed Gene Set Enrichment Analysis (GSEA) [22]. Using the C2 (curated gene sets) collection, we were able to identify several H3K27me3-dependent gene sets among the top ten gene sets to be upregulated in VCaP WT cells (Figure 5b,c). This could be validated by the qRT-PCR of several functional readouts (Figure 5d). In Western blot analysis, we correlated the H3K27me3-dependent gene regulation with a significant upregulation of the histone methyltransferase EZH2 in VCaPrev cells as a possible regulatory mechanism (Figure 5e).

### 3.6. VCaPrev Cells Show a De-Differentiation towards a Neuroendocrine Phenotype Compared to Wildtype VCaP Cells

As the upregulation of EZH2 has shown to be a key hallmark of lineage plasticity and de-differentiation towards a neuroendocrine phenotype [26,27], we decided to further investigate this aspect with regards towards heat sensitivity. Therefore, we re-analyzed the RNAseq data of VCaP and VCaPrev cells using GSEA with the meta-9 dataset defined by Tsai et al. 2017 [28]. In their sequencing meta-analysis, the authors could define gene sets up- and down-regulated in at least 60% of high-grade NEPC patients. Our analysis revealed a NES of 1.04663 comparing VCaPrev and VCaP cells for the upregulated gene sets (Figure 6a). This shows a clear trend; however, the analysis failed to reach significance (FDR 0.397). Regarding the downregulated genes, there was a clearly significant enrichment showing a neuroendocrine de-differentiation of the VCaPrev cells (Figure 6b; NES −2.2607458, FDR < 0.0001). Of note, in this analysis, we could also detect a downregulation of FOLH1, which is known to be a hallmark of neuroendocrine de-differentiation [29]. A previous work of our group identified FOLH1 (or PSMA) downregulation as one of the hallmark characteristics of VCaPrev cells [17]. Investigating other markers of neuroendocrine differentiation, we could also show a downregulation of p53 in VCaPrev cells (Figure 6c).

## 4. Discussion

In the present study, we showed for the first time to our knowledge an in vitro approach for a (focal) pre-treatment enhancing thermosensitivity of PCA cells before HiFU or a similar heat-based approach. Mitoxantrone was shown to significantly increase the amount of apoptotic cells already at 53.9 °C, thereby remaining below the 55 °C postulated before as necessary for induction of cell death [7,10]. In all cell lines investigated in this study besides VCaPrev, metabolic activity commenced decreasing at temperatures around 50 °C and reached close to 0 at 57.8 or 62.6 °C in case of PC3 cells. VCaPrev cells, representing a hormone-sensitive state of PCA and therefore the closest model system to a patient considerable for focal therapy, showed, however, an increased metabolic activity at temperatures between 45.8 and 53.9 °C, which is exactly the temperature range in which the marginal lesion might be during HiFU therapy [30]. PC3 cells were shown to be generally more heat resistant than other cell lines, also indicating their different genotype and phenotype as compared to the rest of PCA cell models.

Supraphysiological temperatures normally induce a heat shock reaction in cells which is largely mediated by seven different classes of proteins among which heat shock proteins are the most important class. The main activator of the heat shock response is the protein HSF1. Under normal conditions, in eukaryotic cells, HSF1 is in a complex with the molecular chaperones Hsp70 and Hsp90. Upon heat stress, the chaperones dissociate from HSF1 which then dimerizes, migrates into the nucleus and acts as a transcription factor [31]. Levels of HSF1 or its chaperones, however, do not change between VCaP and VCaPrev and there is no visible difference in gene expression or protein levels. However, heat response in VCaPrev cells seems to be different from the other cell lines. Its main difference in gene expression is a downregulation of H3K27me3-dependent gene sets (Figure 5b,c). H3K27me3 is a repressive histone modification which is normally written by the histone methyltransferase EZH2, erased by the histone demethylase KDM6A and read by the PRC1 complex. It normally decreases transcriptional activity at its target sites [32].

In our results, we saw an upregulation of EZH2 on protein level in VCaPrev cells (Figure 5e). As EZH2 upregulation is known to be a hallmark of neuroendocrine de-differentation or lineage plasticity, we next investigated such a possible de-differentiation in our cells (Figure 6a–c). Our results clearly show such a resistance mechanism in our cells. This fits with the similar heat response kinetics of PC3 and VCaPrev cells. In addition, PC3 cells show a slight increase in viability at 51.3 °C and a later decrease of viability compared to, e.g., VCaP cells (Figure 1a). PC3 cells are AR and PSA negative and xenograft tumors of PC3 cells are known to form neuroendocrine PCA tumors [33]. This would also explain the difference noted between AR-dependent and AR-independent cell lines noted in our first experiments shown in this manuscript (Figure 1). Inhibition of the AR axis using bicalutamide and finasteride did not lead to any changes in heat response (Figure 2). This indicates that the effect on heat response might not be AR mediated but changes in AR level might only be a side effect of neuroendocrine differentiation. In this model, neuroendocrine PCA would be generally less responsive to focal treatment (e.g., by HiFU). To our knowledge, such aspects have not yet been addressed in the clinic.

Of note, the EZH2 upregulation was not visible on gene expression level (Appendix A). Therefore, we concluded the increase in EZH2 is mediated by post-transcriptional modification, e.g., stabilization. EZH2 is known to be stabilized by Hsp90, which is a heat shock protein [34,35]. It has been previously reported that Hsp90 inhibition can lead to EZH2 destabilization [35].

Therefore, it seems comprehensible that, upon heat induction, Hsp90 also dissociates from EZH2, thereby inducing its degradation and transcriptional de-repression. Furthermore, Hsp90 has been characterized as a general cancer chaperone, not only active in complexing with EZH2 but also with steroid hormone receptors, p53, AKT/PKB or ERBB2 (among others) [36]. Therefore, de-complexion of Hsp90 might need to general transcriptional and metabolic activity. It has also previously been shown by cellular thermal shift assays that 52 °C is the temperature at which EZH2 loses its stability [37]. In our results, we saw a peak in metabolic activity of VCaPrev cells at 51.3 °C, coherent with the aforementioned degradation of EZH2 (Figure 3a).

Another function of Hsp90 is its complexion with type II topoisomerase [38,39]. Type II topoisomerase has been shown to be more active with heat induction, which is why we investigated the effect of mitoxantrone in combination with heat in this study. Topoisomerase II forms a complex with Hsp90, similar to the one formed with EZH2. In MTT assay, we could show that mitoxantrone pre-treatment lead to a significant shift of the heat-induced loss of activity towards lower temperatures (Figure 4a,e) in VCaP cells. However, this result was not reflected in the PI/Annexin V staining (Figure 4g). It has been reported that the inhibition of Hsp90 leads to an increase in in apoptosis caused by topoisomerase II inhibition [38]. Initially, we hypothesized that directly after Hsp90 dissociation, the same effect might be visible. However, PI/Annexin V staining in FACS might not be the adequate method for this setting, as the total cell number differed between treated and untreated cells. For further analysis, we would suggest investigation of other apoptotic assays.

Although we could not show a direct correlation between androgen response and heat response, our findings encourage further study in the differential response of castration-sensitive vs. insensitive tumors (i.e., AR aberration), in heat-based treatments in order to enhance our understanding of treatment failure and make better individual patient-tailored clinical approaches viable.

## 5. Conclusions

Pre-treatment with mitoxantrone in local heat-based treatment options (e.g., HiFU) might be a possible new option which should be further investigated. An effect on heat response seems to correlate with Hsp90-bound complexes and might therefore depend on the levels of its binding partners such as EZH2. Neuroendocrine PCA might be generally more resistant to heat-based treatments. Surely, these results are only in vitro simulations of possible responses and are severely limited by the lack of stromal response or modelling of in vivo mechanisms at the supra-cellular level. Nevertheless, this might suggest a noteworthy approach for enhancing conservative heat-based treatments of PCA and, in the opinion of the authors, warrants further investigation.

## Figures and Tables

**Figure 1 biomedicines-08-00585-f001:**
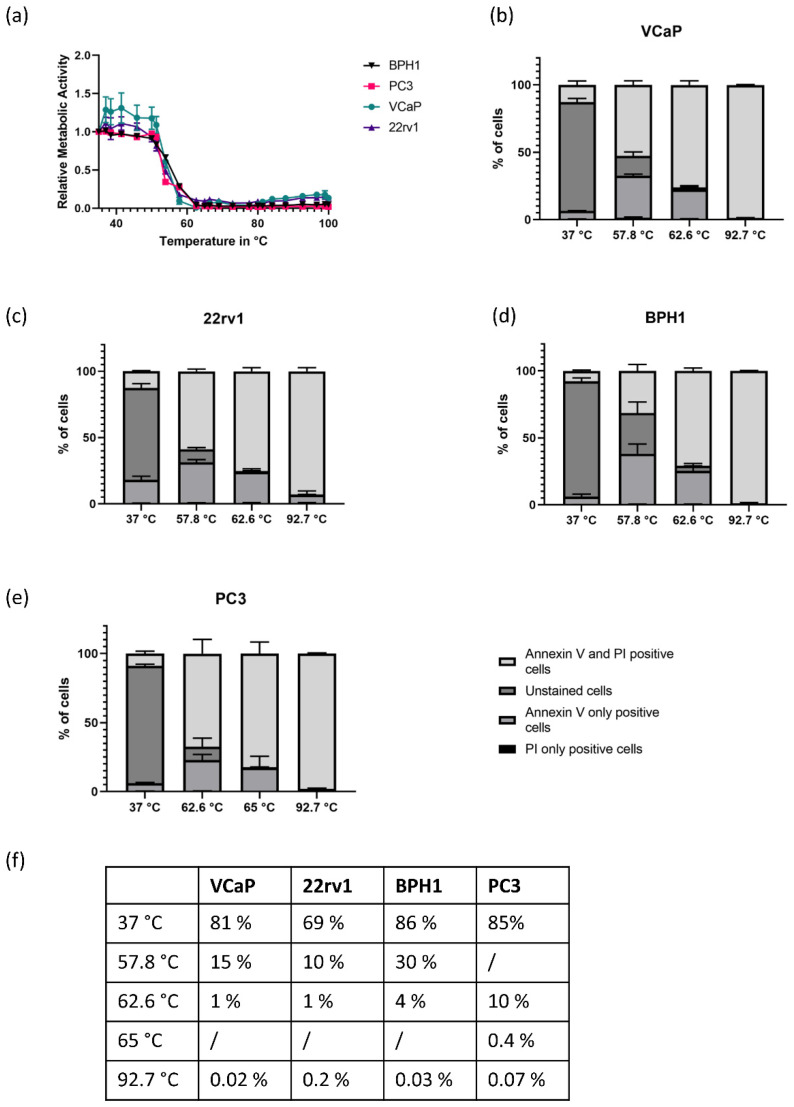
Comparison of different PCA cell lines and their heat response. (**a**) MTT assay of BPH1 and PC3 (AR-negative) as well as VCaP and 22rv1 (AR-positive) cell lines. The results are mean ± standard error of the mean (SEM, shown in error bars) (*n* = 3). (**b**–**e**) FITC PI/Annexin V staining measured by FACS analysis. Temperatures for measuring were defined according to MTT results as the start temperature, the first temperature at which a decrease in metabolic activity was measured, the temperature at which metabolic activity was close to zero and the maximum temperature. Each part of the stacked bar graphs represents one population of cells in red vs. green fluorescence. The results are mean ± SEM (*n* = 3). (**f**) Percentage of viable cells (PI-/Annexin V-) after exposure to different temperatures. At 57.8 °C, a decrease in viability was visible in all cell lines except PC3, where a decrease was visible only at 62.6 °C. The results are mean, *n* = 3.

**Figure 2 biomedicines-08-00585-f002:**
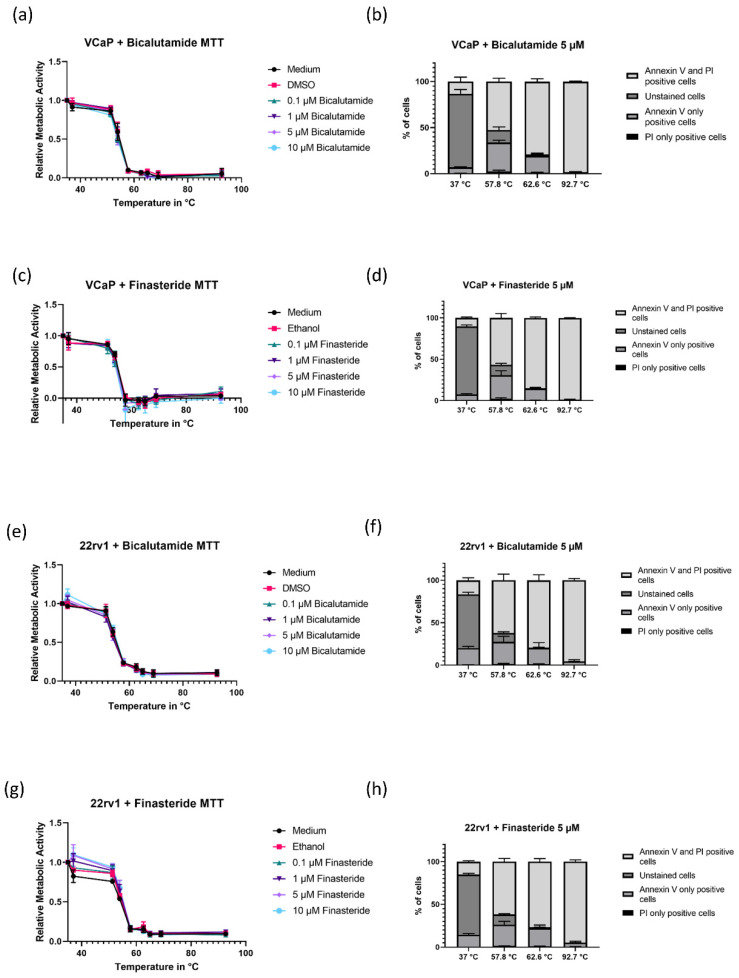
Treatment of AR-positive cell lines with bicalutamide and finasteride and subsequent heat shock. (**a**–**d**) MTT assay and PI/Annexin V staining of VCaP cells with different concentrations of bicalutamide and finasteride. The results are mean ± SEM (*n* = 3). (**e**–**h**) Similar experiment with 22rv1 cells pre-treated with bicalutamide or finasteride.

**Figure 3 biomedicines-08-00585-f003:**
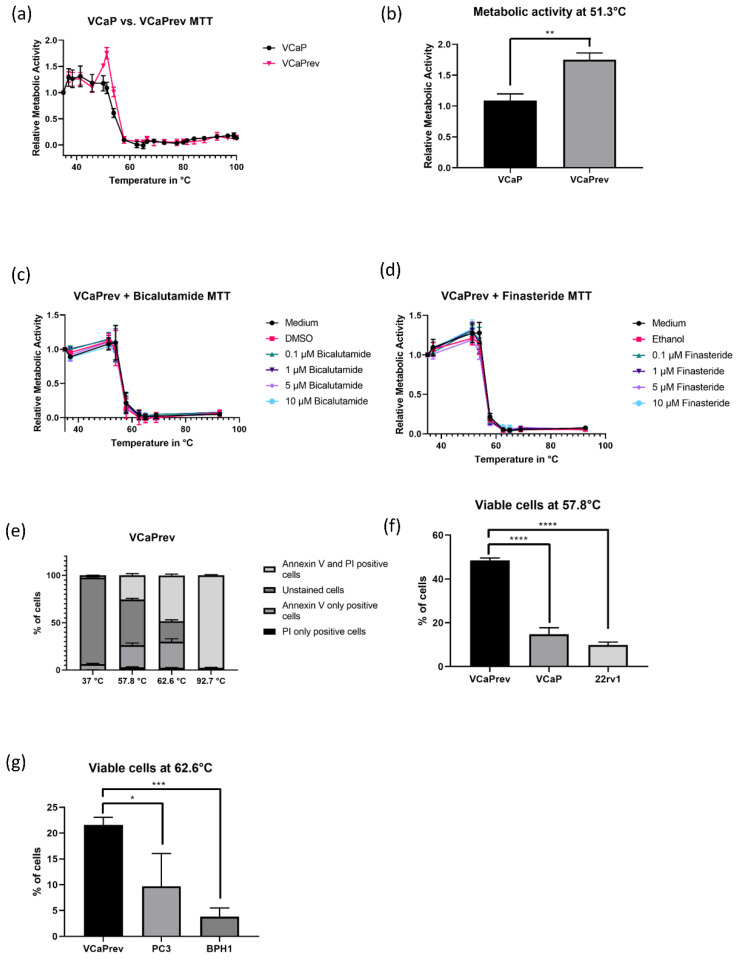
Influence of long-term testosterone treatment on heat sensitivity of VCaP cells. (**a**) MTT assay comparing heat response profiles of VCaP and VCaPrev cells. The results are mean ± SEM (*n* = 3). (**b**) Relative metabolic activity of VCaP and VCaP rev cells at 51.3 °C. (**c**,**d**) Heat response profiles in MTT assay showing pre-treatment with bicalutamide and finasteride. No significant effect is visible. (**e**–**g**) PI/Annexin V staining of VCaP rev cells (results are mean ± SEM, *n* = 3) and comparison of viable cells with other AR-positive cell lines at 57.8 °C and AR-negative cell lines at 62.6 °C. The significance was calculated with unpaired *t*-test, * *p* < 0.05, ** *p* < 0.01, *** *p* < 0.001, **** *p* < 0.0001.

**Figure 4 biomedicines-08-00585-f004:**
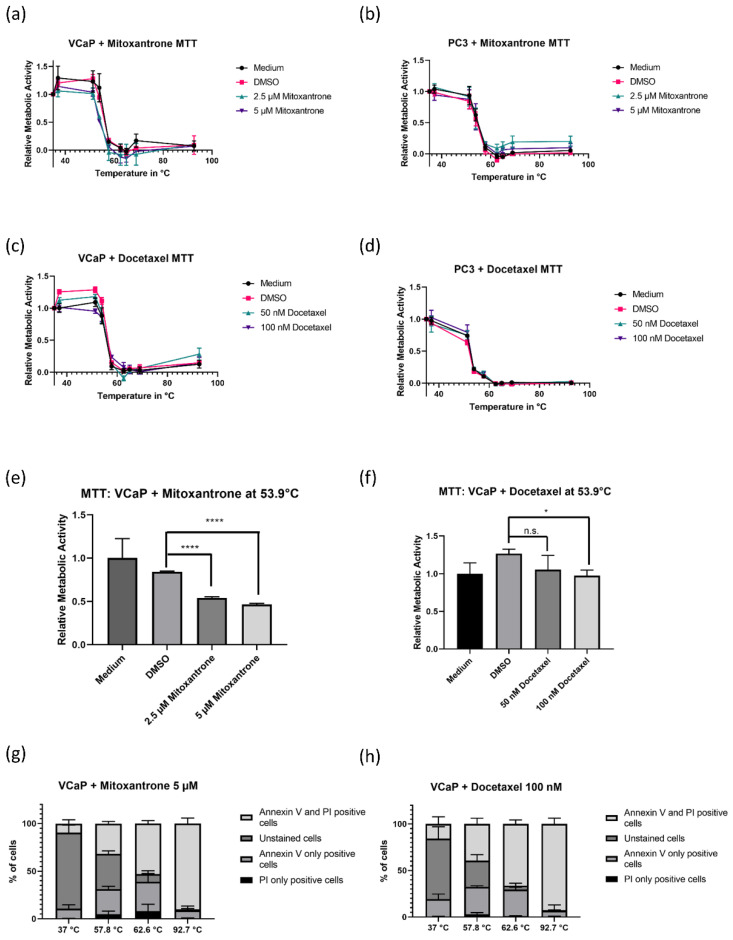
Effect of pre-treatment with type II topoisomerase inhibitor mitoxantrone and control chemotherapeutic docetaxel on heat response. (**a**–**d**) MTT assays comparing heat response after treatment with two different concentrations of mitoxantrone or docetaxel in VCaP and PC3 cells. The results are mean ± SEM (*n* = 3). (**e**) Comparison of relative metabolic activity of VCaP cells with and without mitoxantrone at 53.9 °C. (**f**) Comparison of relative metabolic activity of VCaP cells with and without docetaxel at 53.9 °C. (**g**) PI/Annexin V staining of VCaP cells treated with 5 µM of mitoxantrone before heat shock at different temperatures. (**h**) PI/Annexin V staining of VCaP cells treated with 100 nM of mitoxantrone before heat shock at different temperatures. Significance was calculated with unpaired *t*-test, * *p* < 0.05, **** *p* < 0.0001.

**Figure 5 biomedicines-08-00585-f005:**
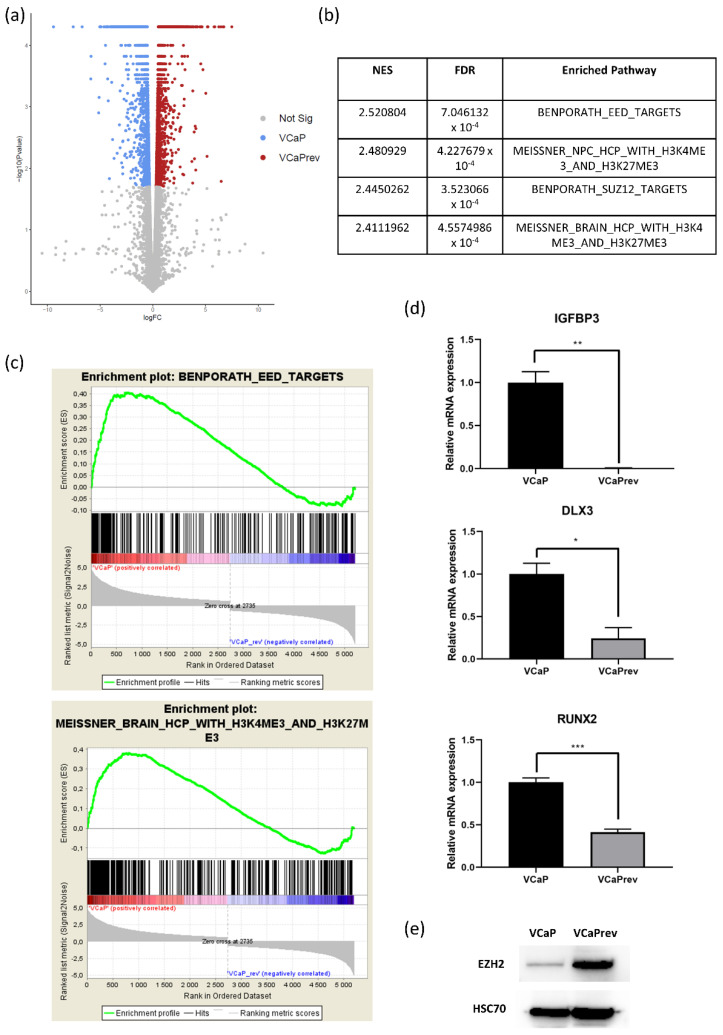
RNAseq analysis of VCaP and VCaP rev cells and target validation. (**a**) Volcano plot showing all differentially regulated genes between VCaP and VCaPrev cells. In red, genes are displayed which are significantly upregulated in VCaPrev cells, whereas blue dots represent genes upregulated in wildtype VCaP. (**b**) H3K27me3 signatures found to be enriched in VCaP cells. Enriched pathways are displayed with normalized enrichment score (NES) and false discovery rate (FDR). (**c**) Enrichment plots of two representative H3K27me3-dependent gene signatures upregulated in VCaP cells. (**d**) Functional validation of H3K27me3-dependent gene regulation at three representative loci present in the before mentioned gene sets by qRT-PCR. The results are mean ± SEM (*n* = 3). Significance was calculated with unpaired *t*-test, * *p* < 0.05, ** *p* < 0.01, *** *p* < 0.001. (**e**) Western blot analysis showing EZH2 regulation in VCaP and VCaPrev cells compared to the loading control HSC70.

**Figure 6 biomedicines-08-00585-f006:**
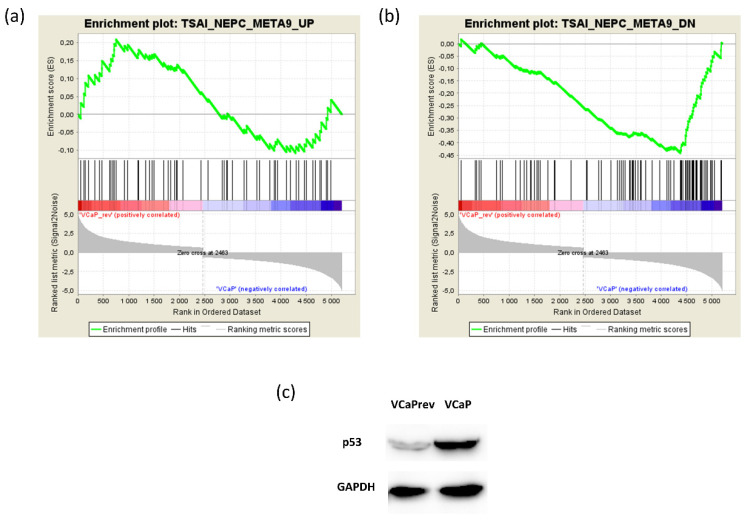
Analysis of a possible neuroendocrine de-differentiation of VCaPrev cells compared to wildtype VCaP cells. (**a**) Enrichment plot of the meta-9 gene set defined by Tsai et al. regarding genes upregulated in neuroendocrine prostate cancer. (**b**) Enrichment plot of the meta-9 gene set defined by Tsai et al. regarding genes downregulated in neuroendocrine prostate cancer. (**c**) Western blot analysis showing p53 regulation in VCaP and VCaPrev cells compared to the loading control GAPDH.

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
