# Peer review of "Modulating the Heat Sensitivity of Prostate Cancer Cell Lines In Vitro: A New Impact for Focal Therapies"

_biomedicines, 2020, doi:10.3390/biomedicines8120585_

Round 1

Reviewer 1 Report

This research paper entitled “Modulating heat sensitivity of prostate cancer cell lines in vitro: a new impact for focal therapies” is clearly presented. The authors have detailed methods and analysis. The discussion section and conclusion paragraph are described completely. This paper can be accepted without revision.

Reviewer 2 Report

The manuscript by Hahn et al. describes the effect of heat on a panel of AR-positive and AR-negative prostate cancer cell lines and also shows that pre-treatment of cells with mitoxantrone can induce an increased heat response. However, results from the study are quite preliminary and do not allow drawing firm conclusions.

Specific comments:

  1. Metabolic activity, as detected by MTT assay, was measured as a correlate of cell viability throughout the study. The assay does not appear appropriate since heat can induce temporary fluctuations in metabolism which have nothing to do with cell viability. Experiments should be repeated with more appropriate assays, such as viable cell counting or, even better, clonogenic assay.
  2. The evidence indicating a correlation between AR and heat response is weak -also by considering that inhibitors of the androgen axis did not modify thermosensitivity of cells- and should be properly discussed.
  3. The relevance of EZH2 and HSP90 for heat resistance should tested by using specific inhibitors.

Reviewer 3 Report

In this manuscript, the effect of elevation of temperature on prostate cancer treatment response has been evaluated using well-stablished cell lines including VCaP, 22Rv1 and PC-3. In addition to these cell lines, a VCaP-driven cell line as a model of high metabolic rates of prostate cancer has been used.

In my review, these this observation is interesting due to its relevance to emerging modalities such as HiFu and nanotechnology-based hyperthermia. Therefore, I recommend this manuscript to be considered for publication after a minor revision. I wish authors address my concern about VCaPrev model.

As you know, EZH2 is elevation is a hallmark of development of NEPC (Beltran et al. 2016 Nature Med). PSMA protein (FOLH1) gene also goes down in NEPC (Bakht et al. 2019 Endocrine-Related Cancer). Suppression of PSMA is associated with elevation of glucose uptake (Bakht et al. 2020 Journal of Nuclear Medicine).    

I noticed that there is a significant suppression in FOLH1 in VCaPrev model (from 160 to 28 RPKM). Also, there is a significant elevation in some of NE markers such as SRRM4 (from 0.9 to 25). In addition, Glucose Transporter 1 (SLC2A1) has an increase. Therefore, it seems VCaPrev is a PSMA-low model with NE-like characters. I recommend author to discuss this point at their discussion or at the results section.   

Round 2

Reviewer 2 Report

The authors addressed the reviewer's comments in the amended version of the manuscript.